# Preoperative Assessment of Perianal Fistulas with Combined Magnetic Resonance and Tridimensional Endoanal Ultrasound: A Prospective Study

**DOI:** 10.3390/diagnostics13172851

**Published:** 2023-09-03

**Authors:** Nikolaos Varsamis, Christoforos Kosmidis, Grigorios Chatzimavroudis, Fani Apostolidou Kiouti, Christoforos Efthymiadis, Vasilis Lalas, Chrysi Maria Mystakidou, Christina Sevva, Konstantinos Papadopoulos, George Anthimidis, Charilaos Koulouris, Alexandros Vasileios Karakousis, Konstantinos Sapalidis, Isaak Kesisoglou

**Affiliations:** 1Third Surgical Department, “AHEPA” University Hospital, Medical Faculty, Aristotle University of Thessaloniki, 1 Kiriakidi Street, 54636 Thessaloniki, Greece; dr.ckosmidis@gmail.com (C.K.); christina.sevva@gmail.com (C.S.); drkonstantinospap1995@gmail.com (K.P.); charilaoskoulouris@gmail.com (C.K.); alexanderkarakousis@gmail.com (A.V.K.); sapalidiskonstantinos@gmail.com (K.S.); ikesis@auth.gr (I.K.); 2Second Surgical Department, “G. Gennimatas” University Hospital, Medical Faculty, Aristotle University of Thessaloniki, 41 Eth. Aminis Steet, 54635 Thessaloniki, Greece; gchatzim@auth.gr; 3Biostatistics Unit, Medical Faculty, Aristotle University of Thessaloniki, 54124 Thessaloniki, Greece; faniapk@gmail.com; 4Surgical Department, ‘St Luke’s’ Hospital, 55236 Thessaloniki, Greece; efthimiadischristoforos@yahoo.gr; 5Radiology Department, “Euromedica” Diagnostic Center, 35 Gr. Lampraki & Ag. Dimitriou Street, 54638 Thessaloniki, Greece; lallasbill9@yahoo.gr; 6Medical Faculty, Aristotle University of Thessaloniki, 54124 Thessaloniki, Greece; chryssa2000@gmail.com; 7Surgical Department, European Interbalkan Medical Center, 10 Asklipiou Street, 55535 Thessaloniki, Greece; anthimid@gmail.com

**Keywords:** perianal fistula, magnetic resonance imaging, tridimensional endoanal ultrasound

## Abstract

Background: we designed a prospective study of diagnostic accuracy that compared pelvic MRI and 3D-EAUS with pelvic MRI alone in the preoperative evaluation and postoperative outcomes of patients with perianal fistulas. Methods: the sample size was 72 patients and this was divided into two imaging groups. MRI alone was performed on the first group. Both MRI and 3D-EAUS were performed in parallel on the second group. Surgical exploration took place after two weeks and was the standard reference. Park’s classification, the presence of a concomitant abscess or a secondary tract, and the location of the internal opening were recorded. All patients were re-evaluated for complete fistula healing and fecal incontinence six months postoperatively. All of the collected data were subjected to statistical analysis. Results: the MRI group included 36 patients with 42 fistulas. The MRI + 3D-EAUS group included 36 patients with 46 fistulas. The adjusted sensitivity and negative predictive value were 1.00 for most fistula types in the group that underwent combined imaging. The adjusted specificity improved for intersphincteric fistulas in the same group. The adjusted balanced accuracy improved for all fistula types except rectovaginal. The combination of imaging methods showed improved diagnostic accuracy only in the detection of a secondary tract. The healing rate at six months was 100%. Fecal incontinence at six months did not present a statistically significant difference between the two groups (Fisher’s exact test *p*-value > 0.9). Patients with complex perianal fistulas had a statistically significant higher probability of undergoing a second surgery (x^2^ test *p*-value = 0.019). Conclusions: the combination of pelvic MRI and 3D-EAUS showed improved metrics of diagnostic accuracy and should be used in the preoperative evaluation of all patients with perianal fistulas, especially those with complex types.

## 1. Introduction

Perianal sepsis presenting as an acute abscess or chronic fistula is a common and benign disease. However, treatment of a perianal fistula may be long and difficult, causing discomfort and anxiety to both the patient and the clinical doctor [1,2].

Surgical treatment of a fistula in ano has two primary goals. The first is to obliterate the internal opening and all the epithelialized tracts and cavities of the fistula. The second is to keep anal continence intact by avoiding injury to the anal muscles [3,4,5]. Westerterp compared these goals with sailing among Scylla and Charybdis [6].

Preoperative imaging of perianal fistulas is crucial, especially in patients whose clinical evaluation demonstrates recurrent or suspected complex fistulas [1,2,5]. Pelvic magnetic resonance imaging (MRI) and tridimensional endoanal ultrasound (3D-EAUS) are the two first-line imaging modalities [1,2,4,7].

In a previous article, we undertook a comprehensive literature review of studies comparing pelvic MRI with EAUS. Many studies were heterogeneous, and their results were contradictory. In most articles, MRI was superior to EAUS in the preoperative evaluation of perianal fistulas. Only in recent articles, with 3D-EAUS and the use of hydrogen peroxide (H_2_O_2_) enhancement, the results of both methods were equal or better for EAUS. The main conclusion was that new, well-designed, and homogenous studies should be conducted, and the combination of pelvic MRI and 3D-EAUS should also be investigated in patients with fistulas-in-ano [2].

Subsequently, we designed an original study that aimed to investigate primarily the diagnostic accuracy of the combination of pelvic MRI and 3D-EAUS versus MRI alone in the detection and anatomical classification of perianal fistulas, preoperatively. A secondary objective was to assess whether the combination of imaging methods is associated with higher rates of perianal fistula healing and preservation of anal continence postoperatively.

We intend to present the results of this study in the article below.

## 2. Materials and Methods

### 2.1. Type of Study

This was a prospective study of diagnostic accuracy that compared the combination of pelvic MRI and 3D-EAUS versus pelvic MRI alone in the preoperative evaluation and postoperative outcomes of patients treated for perianal fistulas. Surgical examination under anesthesia was considered the standard reference. The findings of the imaging methods were correlated with intraoperative findings in all patients. The local ethical committee of the Medical Department of Aristotle University approved the study protocol (IRB protocol number: 2454).

### 2.2. Sample Size

A sample size of 72 patients was calculated to be sufficient to evaluate the two imaging methods, according to the findings of the meta-analysis of Muhammed R. S. Siddiqui et al. [7] and the work of Stark and Zapf [8] on sample size re-estimation, to avoid overfitting in diagnostic accuracy studies.

### 2.3. Patients

The patients in this study came from the anorectal outpatient clinic of the 3rd Surgical Department in “AHEPA” University Hospital, Thessaloniki, from April 2018 to April 2021. During this period, all candidates diagnosed after meticulous clinical examination with a fistula in ano were evaluated for eligibility to participate in the study protocol, and 72 were chosen. All 72 patients provided written informed consent before their inclusion in the study.

### 2.4. Inclusion Criteria

Patients with primary or recurrent perianal fistulas caused by cryptoglandular infection.

### 2.5. Exclusion Criteria

Patients with perianal fistulas associated with:Inflammatory bowel disease (IBD), especially Crohn’s disease;Post-traumatic etiology;Rectal or anal malignancy;Previous pelvic radiotherapy;Immunosuppression;Sexually transmitted anal disease.

### 2.6. Methods

The sample of 72 patients was divided into two imaging control groups. The first group included 36 patients on whom only a pelvic MRI was performed. The second group included 36 patients on whom both pelvic MRI and 3D-EAUS were performed. The findings of the MRI and the 3D-EAUS that were collected included:Anatomical classification of the perianal fistula (Park’s classification) [9];Coexistence of an abscess cavity;Presence of a secondary fistulous tract;Clockwise location of the internal opening in the anal canal.

In some patients, more than one synchronous perianal fistula was detected and recorded separately. In cases of patients with recent surgery for a perianal abscess, the imaging methods were performed one month after abscess drainage and if they visualized a perianal fistula, the patient was enrolled in the study protocol.

Two weeks after the imaging methods were completed, the patients were subjected to the appropriate surgical procedure for the existing perianal fistula. The same four characteristics mentioned above for every fistula in ano found intraoperatively were also recorded. Some patients were admitted to the operating theater more than once to achieve definitive surgical treatment.

Each patient was re-examined with an inspection of the perianal region and digital rectal examination at the anorectal outpatient clinic, six months after the surgical treatment was completed. Two new parameters were evaluated and recorded:Complete healing of the fistula.Presence of fecal incontinence according to the Wexner continence scale (0–20 score) [10].

All of the collected data were finally analyzed with the proper statistical methods.

### 2.7. MRI

Pelvic MRI was performed on a Siemens Avanto 1.5 Tesla magnet scanner. All patients were placed in a supine position and external phased-array body coils were used. Intravenous gadolinium was infused as a contrast medium. No preparation or sedation was utilized.

The study protocol of the MR images acquired in coronal, axial, and sagittal orientations (T1-W FSE, T2-W FSE, STIR) is listed in Appendix A.

Trainees and attending radiologists were involved in the MR imaging. The MR images were finally evaluated by one radiologist with 20 years of experience who was blinded to whether the patient would also be subjected to a 3D-EAUS or not. The anal sphincter muscles, levator plate, and ischiorectal fossa were evaluated with a T1 sequence before and after gadolinium infusion. The lumen of the perianal fistulas and any secondary extensions or inflammatory liquid collections were visualized as hyper-intense tracts or cavities on T2- and STIR-weighted images. On gadolinium-enhanced fat-suppressed T1-weighted images, the fistulous tract’s wall and active granulation tissue demonstrated intense enhancement, while any fluid in the lumen was hypointense. Fibrotic scar tissue demonstrated low signal intensity on both T1- and T2-weighted images. We recorded the four characteristics mentioned in the 2.6. Methods subchapter for all visualized perianal fistulas (Figure 1 and Figure 2).

### 2.8. 3D-EAUS

Evaluation of all patients was performed using a Pro Focus 2202 (BK, Herlev, Denmark) endoanal ultrasound medical device with a model 2052 anorectal transducer. The transducer was equipped with a 360° rotating head that could produce ultrasound waves with a frequency range of 6 to 16 MHz and provide B-mode images. Three-dimensional reconstruction of transversal images in the coronal and sagittal planes was undertaken with the use of specific software (B.K 3D viewer software, v7.0.0.519 (https://bk3d-viewer.software.informer.com/, accessed date 15 August 2023)).

Bowel preparation was performd with a fleet enema 30 min before the examination. The patients were placed in a left-lateral position. The ultrasound probe was covered by a sterile condom, lubricated on both sides, and gently inserted into the upper anal canal until the u-shaped puborectalis muscle was visualized as a hyperechoic band surrounding the posterior wall. The probe was then slowly retracted to visualize the structures of the middle and lower anal canal. The internal anal sphincter (IAS) was recognized as a hypoechoic ring in the upper and middle levels of the anal canal. The external anal sphincter (EAS) was shown as a hyperechoic structure in the outer layer of the middle and lower anal canal.

The 3D-EAUS was performed by a colorectal surgeon certified in endoanal imaging with 10 years of experience who was blinded to the results of the MRI. A colorectal surgeon trainee in endoanal imaging participated in the 3D-EAUS procedure. Hydrogen peroxide enhancement was not applied. A perianal fistula was recognized as a hypoechoic tract that penetrated the anal sphincters or the perianal anatomical spaces and could branch into secondary extensions. The internal orifice of the fistula was shown to disrupt the IAS, according to Cho’s three criteria [11]. A perianal abscess was visualized as a mixed hypo- and hyperechoic region with debris and gas bubbles (Figure 3 and Figure 4).

### 2.9. Surgical Procedure

Examination under anesthesia (EUA) was performed with patients in a high lithotomy position and a laryngeal mask airway (LMA) was used. Bowel preparation was performed with a fleet enema 30 min before surgery. A single dose of Metronidazole 500 mg and Cefoxitin 2 g was administered intraoperatively. The surgical procedure was carried out by two colorectal surgeons with 20 and 10 years of experience, respectively. The findings of the preoperative imaging (MRI alone or the combination of MRI and 3D-EAUS) were studied, and a surgical strategy was planned.

All external orifices of suspected perianal fistulas were examined with pliant fistula probes and an infusion of H_2_O_2_. Characteristics that were examined and recorded for every fistula in ano found included Park’s anatomical classification, the presence of a synchronous abscess cavity or a secondary tract extension, and the clockwise location of the internal opening.

According to Park’s classification, the perianal fistulas were categorized as intersphincteric, transsphincteric, supralevator, and extrasphincteric. Other types of fistulas detected included submucosal, rectovaginal, and blind perineal. All perianal fistulas were also classified as simple or complex, according to the Standard Practice Task Force (2005) and the latest American Society of Colorectal Surgeons (2022) guidelines [2,3,4,5].

Surgical procedures for the treatment of perianal fistulas included the placement of a seton (vessel loop) drain, fistulotomy, Fistula Laser Closure (FiLaC), drainage of a synchronous abscess, unroofing of a secondary tract, and combinations of different methods.

### 2.10. Statistical Analysis

Continuous variables were assessed for normality with the Shapiro–Wilk test and described accordingly with measures of central tendency and variation. The differences between the two groups were assessed with a *t*-test or a Wilcoxon signed rank test, according to normality. Categorical variables were summarized with absolute and relative frequencies, and the asymmetry between groups was assessed with chi-squared, Fisher’s test, or z-test for proportions. The asymmetry between each group with the reference test was assessed with the McNemar–Bowker test of asymmetry. Diagnostic accuracy was assessed with the calculation of sensitivity, specificity, negative predictive value, positive predictive value, negative and positive likelihood ratio, and balanced accuracy. A confusion matrix with the strategy one-vs-rest was used to assess these measures for the primary outcome, since it was a categorical variable with eight levels. To calculate the same measures for the second group (the combination of MRI and 3D-EAUS), a Bayesian approach as described in [12,13] was utilized. It was considered that the two visualizations are connected in parallel and not serially, i.e., both are performed without a time difference that could alter the diagnosis, and the performance of the second does not depend on the result of the first. Significance was held at 5% and power at 80%. Confounding due to predefined parameters was evaluated for secondary outcomes using multinomial logistic regression. All computations were run in the R project for Statistical Computing, v.4.2.1.

## 3. Results

A total of 72 patients with 88 fistulas were screened. The first control group of pelvic MRI included 36 patients with 42 fistulas. The second control group of both pelvic MRI and 3D-EAUS included 36 patients with 46 fistulas. Six patients in the MRI group and ten in the MRI + 3D-EAUS group presented with synchronous fistulas.

The demographic characteristics recorded included race, sex, age, height, and weight, which were used to calculate body mass index (BMI). All 72 patients were Caucasian. There were 55 male patients (76%) and 17 female patients (24%), and the median age was 54 years (interquartile range 45–62) in the total sample. The gender ratio did not differ statistically significantly between the two groups (Pearson’s chi-squared test *p*-value = 0.8), nor did the age (Wilcoxon rank sum test *p*-value > 0.9). Similar results were yielded for height, weight, and BMI. However, it is noteworthy that the median BMI value for the entire sample was 28.6 (SD 25.6–32.0) (Table 1).

Information about previous surgeries in the perianal region was collected. A primary perianal fistula was found in 17 (24%) of patients in the total sample (no statistically significant difference in proportions between groups; Pearson’s chi-squared test *p*-value = 0.8). History of a previous perianal abscess presented equal proportions (50%) in the MRI group, with 69% answering ‘’Yes’’ and 31% ‘’No’’ in the MRI + 3D-EAUS group (Pearson’s chi-squared test *p*-value = 0.093). History of fistula recurrence did not show a statistically significant difference between the two groups (Pearson’s chi-squared test *p*-value = 0.4). Five patients (14%) in the MRI group and eight patients (22%) in the MRI + 3D-EAUS group presented with recurrent perianal fistulas (Table 1).

A statistically significant difference was observed in the ratio of simultaneous treatment for perianal abscess and fistula between the groups (Fisher’s exact test, *p* = 0.003). Twelve patients (33%) in the MRI group and two patients (5.6%) in the MRI + 3D-EAUS group underwent synchronous surgery for abscess and fistula. Classification of perianal fistulas into simple and complex types did not have a statistically significant difference between the two imaging control groups (Pearson’s chi-squared test *p*-value = 0.8). In the MRI group, 19 patients (53%) presented with simple and 17 patients (47%) with complex perianal fistulas. In the MRI + 3D-EAUS group, 20 patients (56%) were diagnosed with simple and 16 patients (44%) with complex fistulas (Table 1).

Treatment of the disease with seton placement was the most common surgical procedure in the total sample of perianal fistulas (60.2%) and in both imaging control groups (MRI: 54.8%, MRI + 3D-EAUS: 65.2%). Fistulotomy was the second most common procedure, with a total percentage of 27.3% (MRI: 28.5%, MRI + 3D-EAUS: 26%). The two most regularly performed supplementary surgical procedures were perianal abscess drainage (total sample of perianal fistulas: 16%, MRI: 28.5%, MRI + 3D-EAUS 4.3%), and unroofing of a secondary fistulous tract (total sample of perianal fistulas: 14.8%, MRI: 7.1%, MRI + 3D-EAUS 21.7%).

Sixteen patients in the MRI group were treated with combined surgical procedures, and twelve (33%) of them underwent seton placement along with perianal abscess drainage. In the MRI + 3D-EAUS group, 10 patients (27.8%) were treated with both seton placement and unroofing of a secondary tract out of 15 patients that underwent combined surgical procedures.

Overall, 49% of patients (MRI 42%, MRI + 3D-EAUS 56%) underwent a second surgical procedure for definitive treatment of the disease. The most frequent repeat operation was seton suture change (MRI 72.7%, MRI + 3D-EAUS 86.7%).

### 3.1. Park’s Classification of Fistulas

Assuming that the discovery of a second fistula in the same patient is independent of the coexisting fistula, the frequencies of fistula types for the preoperative MRI-only group, combined imaging, and intraoperative classification are shown in Table 2. The global symmetry test revealed no statistically significant difference between the groups, both initially and after correction for multiple comparisons. It was not possible to calculate sensitivity and specificity for supralevator fistulas in the MRI group, as there was no observation. There were a few extrasphincteric, rectovaginal, and perineal fistulas in each group.

The calculated diagnostic accuracy metrics are shown in Table 3. Notably, sensitivity improved in the MRI + 3D-EAUS group for transsphincteric fistulas and submucosal fistulas. Specificity improved considerably for intersphincteric and slightly for extraspincteric fistulas. The negative predictive value improved in most categories except for submucosal and rectovaginal fistulas. Balanced accuracy improved in all types of fistulas except for rectovaginal.

All values of balanced accuracy for each type of fistula were compared with the z-test for proportions in both imaging control groups. A statistically significant (*p* < 0.05) difference was found in the intersphincteric, submucosal, rectovaginal, and no-finding categories. The combination of imaging methods showed a statistically significant higher balanced accuracy for the intersphincteric, submucosal, and no-finding categories, whereas MRI alone showed statistically significantly better results for rectovaginal fistulas.

#### 3.1.1. Transsphincteric Fistulas

The diagnostic accuracy of the test combination improved in terms of sensitivity for the detection of transsphincteric fistulas, with a corresponding loss of specificity. The negative predictive value increased in the combined test, from 0.78 in the MRI group to 1.00 in the MRI + 3D-EAUS group, meaning that 100% of patients negative for transsphincteric fistula on either MRI, 3D-EAUS, or both, would indeed not suffer them. The positive predictive value decreased from 0.89 to 0.71, meaning that 71% of patients with a positive MRI, 3D-EAUS, or both, would have a transsphincteric fistula. The LR+ positive likelihood ratio, i.e., the probability of the presence of disease in patients with at least one positive test, was 2.67.

#### 3.1.2. Intersphincteric Fistulas

The diagnostic accuracy of the test combination improved in terms of specificity for the detection of intersphincteric fistulas, with a loss of sensitivity from 1.00 to 0.89. The negative predictive value remained the same on combined imaging, at the 1.00 level, thus 100% of patients with negative results from either imaging method would not have an intersphincteric fistula. The positive predictive value remained the same, from 0.68 to 0.67, i.e., 67% of patients with at least one positive imaging method would have an intersphincteric fistula. The positive likelihood ratio LR+, i.e., the probability of disease presence in patients with a positive in at least one test, was 13.3.

#### 3.1.3. Supralevator Fistulas

No supralevator fistulas were diagnosed in the MRI group. The imaging combination had a combined sensitivity of 1.00 and specificity of 1.00. The negative predictive value was 1.00, and the positive predictive value was 1.00. Therefore, 100% of patients with at least one negative test would not have a supralevator fistula, while 100% of patients with at least one positive test would. This result should be interpreted with great caution, as the relevant table includes only three diagnosed fistulas, and no positive examination was refuted intraoperatively. The negative likelihood ratio LR-, i.e., the probability of absence of disease in patients with a positive result on at least one test, was 0.25.

#### 3.1.4. Extrasphincteric Fistulas

The sensitivity and specificity of extrasphincteric fistula diagnosis by MRI were 1.00 and 0.95, respectively. For the MRI + 3D-EAUS group, the specificity and sensitivity were 0.98 and 1.00. The negative predictive value increased to 1.00 in the combined imaging group, thus 100% of patients with any negative imaging will not have an extrasphincteric fistula. The corresponding value was 0.98 for the MRI group. The positive predictive value increased from 0.33 to 0.50 in the combined imaging group.

#### 3.1.5. Submucosal Fistulas

The sensitivity of diagnosing submucosal fistulas improved in the imaging combination, from 0.40 to 0.63, while the specificity decreased from 1.00 to 0.92. The negative predictive value increased from 0.92 in the MRI group to 0.95 in the combined imaging group, meaning that for 95% of patients with even one negative test, a submucosal fistula would not be found. The positive predictive value decreased from 1.00 to 0.71, meaning that 71% of patients with at least one positive test would have a submucosal fistula. The positive likelihood ratio LR+, i.e., the probability of disease presence in patients with a positive result on at least one test, was 14.3.

#### 3.1.6. Rectovaginal Fistulas

Only one rectovaginal fistula was diagnosed in the MRI group, which had been visualized preoperatively with MRI. Only two rectovaginal fistulas were diagnosed in the MRI + 3D-EAUS group, of which only one was detected by 3D-EAUS and none by MRI (one was diagnosed as anterior transsphincteric). The calculated diagnostic accuracy values are not reliable for this type of fistula.

#### 3.1.7. Perineal Fistulas

Only one perineal fistula was diagnosed in the MRI group, which had been imaged preoperatively with an MRI. Only two perineal fistulas were diagnosed in the MRI + 3D-EAUS group, both of which were detected by MRI and one by 3D-EAUS. Estimated diagnostic accuracy values are not reliable for this type of fistula.

#### 3.1.8. No Findings

Preoperative absence of imaging findings had a negative predictive value of 0.92 in the MRI group and 0.97 in the MRI + 3D-EAUS group. Thirteen percent of patients with at least one positive imaging in the MRI + 3D-EAUS group would remain without findings at surgical exploration (positive predictive value 0.13). Pooled sensitivity was 0.67, compared to null in the MRI group, and specificity was 0.72 versus 1.00.

### 3.2. Presence of a Concomitant Abscess Cavity

Thirty-six patients were screened in each imaging control group, with 42 fistulas found in the first group (MRI) and 46 in the second group (MRI + 3D-EAUS).

The presence of a synchronous abscess cavity was checked in both imaging groups, and it was shown that in the MRI group, 16 patients had a concomitant abscess cavity and 25 did not, compared with the intraoperative detection of 15 abscess cavities (Table 4). Twelve of these fifteen patients had an acute abscess and underwent synchronous treatment for both the abscess and the perianal fistula (Table 1). The remaining three patients had a residual abscess cavity after previous surgical drainage and were treated only for the perianal fistula.

In the MRI + 3D-EAUS group, six patients had a concomitant abscess cavity on preoperative MRI imaging and five on preoperative 3D-EAUS. Eight patients had a confirmed abscess cavity intraoperatively (Table 4). Two of these eight patients had an acute perianal abscess (Table 1), and the other six had a residual abscess cavity.

The specificity and sensitivity for the first group (MRI) were 0.96 and 1.00, respectively. Diagnostic accuracy was found to be 0.975, with 95% CI: 0.868, 0.999. The *p*-value (McNemar’s test) was found to be almost one. The negative predictive value was 1.00, meaning 100% of patients with a negative MRI did not have a synchronous abscess cavity, and the positive predictive value was found to be 0.94, meaning 94% of patients with a positive MRI did have a concomitant abscess cavity.

In the MRI + 3D-EAUS group, the combined specificity was 1.00 and the combined sensitivity was 0.91. The negative predictive value was 0.97, i.e., 97% of patients with at least one negative test did not have an accompanying abscess cavity. The positive predictive value was 1.00; therefore, 100% of patients with at least one positive imaging will be found to have an accompanying abscess cavity. The negative likelihood ratio was 0.125.

### 3.3. Presence of a Secondary Fistulous Tract

The existence of a secondary fistulous tract in the MRI-only group was detected in five patients preoperatively and confirmed in three cases intraoperatively. In the combined imaging group, thirteen patients had a secondary fistulous tract identified on preoperative MRI, seven on 3D-EAUS, and ten had an intraoperative finding of a secondary tract (Table 5).

The specificity and sensitivity for the first group (MRI) were 0.92 and 0.34, respectively. Diagnostic accuracy was found to be 0.875, with 95% CI: 0.732, 0.958. The *p*-value (McNemar’s test) was found to be almost one. The negative predictive value was 0.94, meaning 94% of patients with a negative MRI did not have a secondary fistulous tract, and the positive predictive value was found to be 0.25, meaning 25% of patients with a positive MRI did have a secondary fistula.

In the MRI + 3D-EAUS group, the combined specificity was 0.84 and the combined sensitivity was 0.91. The negative predictive value was 1.00, i.e., 100% of patients with at least one negative test did not have a secondary fistulous tract. The positive predictive value was 0.7; 71% of patients with at least one positive imaging had a secondary fistula. The positive likelihood ratio was 7.14.

### 3.4. Inner Orifice Location

Data were recorded at each finding site, clockwise, to locate the internal orifice. As seen from Table 6, most fistulas were detected on the middle-posterior wall of the anal canal (6 o’clock). For this reason, the remaining positions were collapsed into one category, “Other location”, and the absence of an internal orifice remained in a separate category.

The specificity and sensitivity for locating the inner orifice at 6 o’clock were 0.97 and 1.00 in the MRI group, and the combined specificity and sensitivity in the MRI + 3D-EAUS group were 0.95 and 1.00, respectively. The negative predictive value of combined imaging increased to 1.00 (that of single MRI localization was found to be 0.88). The positive predictive value was reduced to 95% versus 100% for imaging alone.

At the other sites, the sensitivity and specificity were 0.92 and 0.93 for the MRI group and 1.00 and 0.89 for the MRI + 3D-EAUS group, respectively. The negative predictive value for combined imaging increased to 100% from 96% in the MRI group, and the positive predictive value increased to 90% from 86%.

Absence of internal orifice was detected with a sensitivity of 1.00 and specificity of 0.97 in the MRI group and 1.00 and 0.81 in the MRI + 3D-EAUS group. The positive predictive value was 0.67 in the MRI group and 0.13 in the MRI + 3D-EAUS group, and the negative predictive value remained 100% in both groups.

### 3.5. Perianal Fistula Healing and Fecal Incontinence Six Months Postoperatively

The relative frequencies for each of the secondary outcomes of interest are shown in Table 7. A comparison of the perianal fistulas’ healing rates was not possible, since healing was achieved in all patients at the 6-month follow-up, which was the prespecified outcome of interest. Follow-up was continued for six more months, and two fistula recurrences were detected, one in each imaging control group.

Regarding fecal incontinence, three patients in the MRI group and two in the MRI + 3D-EAUS group had mild incontinence according to the Wexner scale criteria, and the rest had no discomfort. The comparison with Fisher’s exact test for the group-wise correlation matrix had a *p*-value > 0.9, i.e., it was not statistically significant (Table 7).

The repeat surgery rate for definitive treatment did not show a statistically significant difference in proportions between the two imaging control groups (Pearson’s chi-squared test *p*-value = 0.2) (Table 7).

The sample of 72 patients was also divided into a group with simple perianal fistulas (39 patients) and a group with complex perianal fistulas (33 patients). The healing rate at six months postoperatively was equal (100%) in patients of both groups (Table 8). Two patients were diagnosed with perianal fistula recurrence, one in the group with simple and one in the group with complex fistulas, at one year of follow-up. This was outside the prespecified period for our study.

Fecal incontinence, according to the Wexner score, did not present a statistically significant difference between the two groups (Fisher’s exact test *p*-value > 0.9). Most patients (93% in the total sample, 92% with simple fistula, 94% with complex fistula) had normal anal continence at the sixth postoperative month (Table 8).

Patients with complex perianal fistulas had a 3.2 times higher probability of undergoing repeat surgery for definitive treatment of the disease compared to patients with simple fistulas (OR 3. 13, 95% CI 1.19–8.2, *p*-value 0.019) (Table 8).

## 4. Discussion

The incidence of perianal fistulas in Europe ranges from 1.7 to 2.3 cases per 10,000 persons/year, and the total prevalence in Europe is 1.69 per 10,000 population. The male-to-female ratio is approximately 2/1, and the mean age of patients is 40 years [2,14,15]. In our study, the male-to-female ratio was approximately 3/1, and the mean age was higher (54 years) for the total sample of patients (Table 1). Obesity (BMI > 25 kg/m^2^) is a recognized independent risk factor for perianal fistula formation [2,16]. This is in concordance with our study, where the median BMI value for the total sample was 28.6 kg/m^2^ (Table 1).

Pathogenesis of perianal fistulas is mainly associated with cryptoglandular infection, and Crohn’s disease is a second, rarer cause [2,17]. Most published articles that compare MRI with EAUS are heterogeneous and include perianal fistulas with different pathogenesis [1,2,18,19]. In our study, we only included patients with perianal fistulas of cryptoglandular origin, which is the most common cause of the disease. Thus, our sample is homogenous, and the results of the study are more reliable and reproducible.

According to the theory of cryptoglandular infection, 40–60% of patients treated for an acute perianal abscess will eventually develop a chronic fistula-in-ano. Previous anorectal surgery is a second independent risk factor for fistula formation and recurrence. [2,16,17,20]. The incidence of recurrence depends on the type of fistula and is highly variable, ranging between 7% and 50% [21,22]. Maier et al. detected no statistically significant correlation between the patient’s history (Crohn’s disease or recurrent fistula) and the diagnostic accuracy of MRI and EAUS [18]. In our study, 43 (60%) out of 72 patients had a history of acute perianal abscess drainage, and 13 (18%) out of 72 patients presented with recurrent perianal fistula (Table 1). Only 15 (21%) of our total patients had a history of no previous anorectal surgery.

Pelvic MRI is considered the gold-standard method of imaging for perianal fistulas. It is related to statistically significant better results and prognosis after surgical treatment of fistulas-in-ano [2]. It can offer more information than examination under anesthesia, alter the treatment strategy, and reduce postoperative recurrence rates of the disease [23,24,25,26,27,28,29]. The 3D-EAUS method has high diagnostic accuracy in detecting the internal opening of a fistula in-ano, as well as intersphincteric and transsphincteric fistulas [2,7]. It can reliably facilitate surgical planning, identify defects of the anal sphincters, and it is associated with statistically significant better results in preserving anal continence postoperatively, especially in complex cases [30,31,32].

Shwartz et al. compared the diagnostic accuracy of pelvic MRI, endoanal ultrasound, and examination under anesthesia in 32 patients with Crohn’s disease. Thirty-nine peri-anal fistulas and 13 abscesses were detected. The diagnostic accuracy when any two of the three mentioned tests were combined was 100% [33]. Other published articles also suggested that if pelvic MRI and EAUS were combined preoperatively, this could improve their total diagnostic accuracy [34,35].

To the best of our knowledge, our article is the first study in the literature to calculate measures of diagnostic accuracy for the combination of pelvic MRI and 3D-EAUS compared to pelvic MRI alone. According to our results, the combination of the two imaging methods presented an adjusted sensitivity and an adjusted negative predictive value of 1.00 in all major categories of perianal fistulas (transsphincteric, intersphincteric, supralevator, extrasphincteric, and blind perineal). The adjusted sensitivity also improved for submucosal fistulas (0.71 from 0.40) and the no finding category (0.67 from 0.00). The adjusted negative predictive value decreased from 1.00 to 0.95 for submucosal and increased from 0.92 to 0.97 for the no finding category (Table 3).

The adjusted specificity improved for intersphincteric (0.93 from 0.66) and extrasphincteric (0.98 from 0.95) fistulas, and was 1.00 for supralevator, rectovaginal, and blind perineal fistulas. The adjusted specificity decreased for transsphincteric (0.63 from 0.96) and submucosal (0.95 from 1.00) fistulas and for the no finding category (0.70 from 1.00). The adjusted positive predictive value increased for intersphincteric (0.67 from 0.33), extrasphincteric (0.50 from 0.33), and rectovaginal (1.00 from 0.68) fistulas, and was 1.00 for supralevator and blind perineal fistulas. The adjusted positive predictive value decreased for transsphincteric (0.71 from 0.89) and submucosal (0.71 from 1.00) fistulas, and the no finding category (0.13 from 1.00) (Table 3).

The only meta-analysis in the literature that compared the diagnostic accuracy of pelvic MRI to EAUS in the preoperative imaging of perianal fistulas showed equal combined sensitivity for both methods, with a value of 0.87 and a comparable range. Pelvic MRI presented a combined specificity rate of 0.69 and EAUS a lower rate of 0.43 [7]. This meta-analysis included only four articles for quantitative synthesis, with a high degree of heterogeneity for reporting sensitivity and other measures of diagnostic accuracy [18,19,36,37]. In more recently published articles, the results were also contradictory. Brillantino et al. reported an equal value of sensitivity for MRI and peroxide-enhanced 3D-EAUS (98.4%), and an equal value of specificity for both imaging methods (83.3%) [1]. Akhoundi et al. calculated the following measures of diagnostic accuracy: MRI sensitivity = 76.12%, EAUS sensitivity = 87.38%, MRI specificity = 57.69%, EAUS specificity = 38.46%, MRI PPV = 93.88%, EAUS PPV = 92.38%, MRI NPV = 22.05%, EAUS NPV = 26.31%, MRI accuracy = 74.19%, and EAUS accuracy = 82.25% [38]. In comparison to most of the previous articles, our study presented highly improved measures of diagnostic accuracy regarding the combination of pelvic MRI and 3D-EAUS.

According to the literature, pelvic MRI has a high tendency to detect a secondary fistulous tract and a concomitant abscess cavity. Endoanal ultrasound tends toward a more accurate detection of the internal opening and relationship with the anal sphincters, due to their proximity to the probe [2,4,7]. Buchanan et al. found a statistically significant higher accuracy of MRI in detecting primary tracts (*p* < 0.001), horseshoe extensions (*p* = 0.003), abscess cavities (*p* < 0.001), and internal openings (*p* < 0.001) in comparison with EAUS and clinical examination. Nevertheless, they considered EAUS reliable for the identification of the inner orifice of a perianal fistula [19]. Brillantino et al. reported the overall sensitivity of MRI and 3D-EAUS at 94.4 and 80.6, respectively, in the detection of secondary extensions for simple and complex fistulas. The overall specificity of MRI and 3D-EAUS was 98.9 and 97.8, respectively, in the same category. However, MRI was superior to 3D-EAUS for the detection of secondary extensions in complex perianal fistulas (*p* = 0.041 McNemar’s test). The overall sensitivity of MRI and 3D-EAUS in the identification of the internal opening was 95.8% and 97.5%, respectively. The overall specificity had an equal value of 83.3% for MRI and 3D-EAUS in the same category [1]. Sayed et al. also concluded that MRI was superior to 3D-EAUS for the evaluation of secondary tracts (K-value = 0.65, *p* < 0.001). Ultrasound was found to be superior to MRI in the localization of the internal opening (K-value = 0.44, *p* <0.001) [39]. Gustafson et al. presented the only study that presented better results for EAUS with probing than pelvic MRI regarding secondary characteristics of perianal fistulas [36]. In our study, the combination of pelvic MRI and 3D-EAUS showed improved sensitivity, and negative and positive predictive value only in the detection of a secondary extension, whereas metrics for a synchronous abscess and the internal opening were almost equal between the two imaging control groups.

Surgical treatment of a perianal fistula is considered successful if complete healing of the wound is achieved within six months after the final surgical procedure. If the healing is not completed within six months and an external opening with fluid discharge remains, the perianal fistula is defined as persistent. Recurrence of a perianal fistula occurs within one year after definitive surgical treatment. It is defined as a clinical relapse of the disease after complete healing of the surgical wound. In cases where the clinical relapse occurs later than one year, the disease is defined as a de novo perianal fistula [22]. In our study, we investigated the complete healing of perianal fistulas in all patients, and therefore we set the endpoint of follow-up at six months after definitive surgical treatment. The recorded rate of complete healing was 100% in all patients in both imaging control groups. Nevertheless, two patients (one from each imaging control group) were re-admitted with relapses of the disease within one year after the completion of the surgical treatment. They were considered recurrent perianal fistulas outside the prespecified period of follow-up.

Surgical treatment of a perianal fistula is also considered successful if anal continence is preserved intact postoperatively [3,4,5]. There are many scales of fecal incontinence described in the literature that validate the severity of different symptoms and their effect on quality of life. In our study, we used the Wexner/Cleveland Clinic Fecal Incontinence Score, which is simple and can easily be applied to a patient. It evaluates five parameters (solid stool, liquid stool or gas incontinence, usage of pads, lifestyle alteration), and each parameter is graded on a scale from zero to four, with a high score of twenty. The Wexner score is one of the most widely adopted fecal incontinence scales, which can be utilized in clinical practice and for research purposes [10,40].

Abbas et al. studied 179 patients treated for perianal fistulas of cryptoglandular origin. They reported baseline stool or gas incontinence symptoms in 7.3% of the patients postoperatively [41]. Van Koperen et al. also studied 179 patients treated for perianal fistulas of cryptoglandular origin. One hundred and nine (109) patients were treated for low fistulas and 70 patients were treated for high fistulas. A significant percentage (40%) of patients in both groups reported soiling postoperatively [42]. Jayarajah et al. studied 34 patients with simple cryptoglandular fistula in ano who were treated with surgery. The overall preoperative and postoperative rates of incontinence were 18% and 38%, respectively [43]. In our study, five patients (7%) in the total sample had symptoms of mild incontinence postoperatively. Three patients (4.2%) had a Wexner score of three (two in the MRI group, one in the MRI + 3D-EAUS group), and two patients (2.8%) had a Wexner score of four (one in each imaging control group). There was no statistically significant difference in fecal incontinence postoperatively between the two groups (Table 7 and Table 8).

Finally, complex perianal fistulas are considered a distinct category with challenging surgical treatment [44]. Patients with Crohn’s disease may require up to six surgical procedures for complex fistulas [45]. Burney published a case-series study of 483 patients with perianal fistulas that reported the long-term results of surgical management. Patients with complex and recurrent perianal fistulas (18%) had to undergo up to four surgical operations before complete healing was achieved [46]. In our study, patients with complex perianal fistulas also had a higher probability of undergoing more than one surgical procedure to eliminate the disease compared to patients with simple fistulas.

### Study Limitations

This was a single-center study with few participants. The imaging modalities of MRI and 3D-EAUS were both assessed by a single experienced and certified evaluator, which may reduce the reliability of the diagnostic results. Similar future studies should be multicentric and enroll two or more independent evaluators for each imaging modality.

The categories of supralevator, extrasphincteric, rectovaginal, and perineal fistulas, as well as the no finding category, included a relatively small number of patients. The incidence of these types of perianal fistulas was also low in recently published studies (supralevator 9.5%, extrasphincteric 0%, rectovaginal 5%) [5,47,48]. Nevertheless, the values of diagnostic accuracy for the above fistulas in our study should be interpreted with great caution. Future studies of combined diagnostic accuracy for pelvic MRI and 3D-EAUS should focus on intersphincteric and transsphincteric perianal fistulas, which are the two prevalent types, or include an increased number of patients for the other less common categories.

The postoperative follow-up for our patients was defined as six months. However, two cases of perianal fistula recurrence occurred later, within the first year after surgical treatment. Future studies that will record postoperative rates of perianal fistula healing and recurrence should have a follow-up period of at least 12 months. They could also use a more objective scoring system, which would evaluate post-surgery healing and predict long-term results better than clinical examination alone. Garg et al. published an article in 2022 that describes such a novel scoring system with six weighted parameters (0–20 score). Four parameters were based on MRI performed at least 3 months postoperatively and two parameters were clinical. This article was published after the protocol of our study was completed [49].

## 5. Conclusions

The combination of pelvic MRI and 3D-EAUS has higher diagnostic accuracy than MRI alone in the detection and anatomical classification of all types of perianal fistulas preoperatively, except for rectovaginal fistulas, which are better visualized with pelvic MRI alone. The diagnostic accuracy in the detection of a concomitant abscess cavity and the internal opening of a perianal fistula is similar between pelvic MRI alone and the combination of MRI and 3D-EAUS. However, the combination of imaging methods is associated with improved measures of diagnostic accuracy in the visualization of a secondary fistulous tract.

The combination of imaging methods is not associated with statistically significant higher rates of perianal fistula healing and lower rates of fecal incontinence at six months postoperatively. Patients with complex perianal fistulas have a statistically significant (*p*-value = 0.019) higher probability of undergoing repeat surgery for definitive treatment of the disease compared to patients with simple fistulas.

Based on the results of our study, we strongly recommend that both pelvic MRI and 3D-EAUS be used in the preoperative evaluation of all patients with perianal fistulas, especially those with complex types.

## Figures and Tables

**Figure 1 diagnostics-13-02851-f001:**
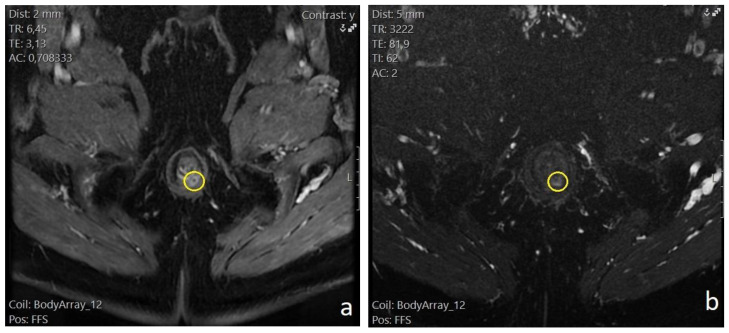
Patient no. 1; preoperative pelvic MRI. (**a**) T1 fat-saturated sequence with intravenous contrast, axial plane. Intersphincteric perianal fistula, located at the posterior wall (5 o’clock) of the anal canal (yellow circle). Intense enhancement of the fistulous tract wall. (**b**) T2 fat-sat, axial plane. The same 5 o’clock intersphincteric perianal fistula (yellow circle). Hyperechoic visualization of the fistula lumen.

**Figure 2 diagnostics-13-02851-f002:**
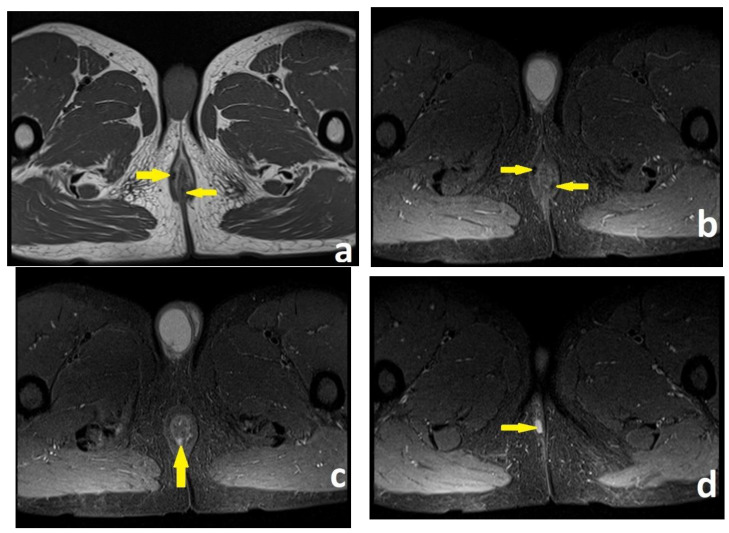
Patient no. 2; preoperative pelvic MRI. (**a**) Axial T2-weighted sequence. Intersphincteric perianal fistula, located at the 11 o’clock in the anal canal (left yellow arrow). Submucosal perianal fistula, located at the 6 o’clock in the anal canal (right yellow arrow). (**b**) Axial T2 fat-saturated sequence. The same two perianal fistulas (yellow arrows). (**c**) Axial T2 fat-sat sequence. Submucosal perianal fistula at the posterior wall (6 o’clock) of the anal canal (yellow arrow). (**d**) Axial T2 fat-sat sequence. Concomitant small abscess at 11 o’clock (yellow arrow).

**Figure 3 diagnostics-13-02851-f003:**
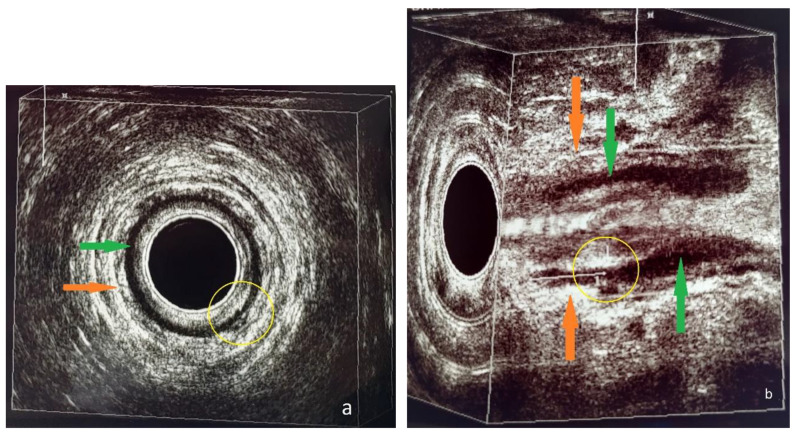
Patient no. 1; preoperative 3D-EAUS. (**a**) Axial plane. Intersphincteric perianal fistula with an internal opening at the posterior wall (5 o’clock) of the anal canal (yellow circle), according to Cho’s criterion I [11]. Internal anal sphincter, visualized as a black, hypoechoic ring (green arrow). External anal sphincter, visualized as a bright, hyperechoic circular structure (orange arrow). (**b**) Sagittal plane, tridimensional reconstruction. Intersphincteric perianal fistula with an internal opening at the posterior wall (5 o’clock) of the anal canal at a depth of 17 mm from the anal verge (yellow circle). Internal anal sphincter, visualized as two black, hypoechoic lines (green arrows). External anal sphincter, visualized as a bright, hyperechoic circular structure (orange arrows).

**Figure 4 diagnostics-13-02851-f004:**
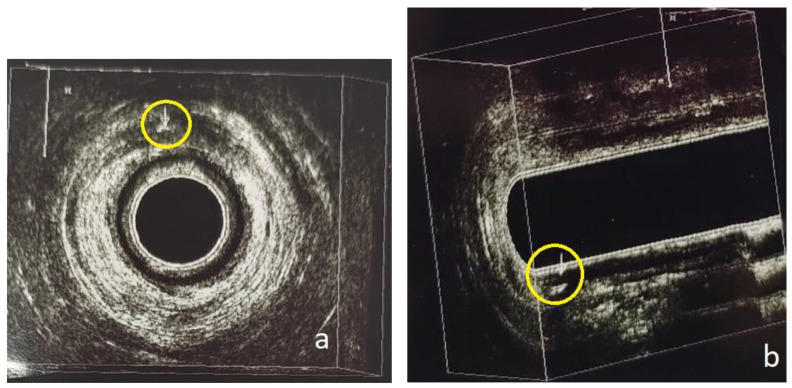
Patient no. 2; preoperative 3D-EAUS. (**a**) Axial plane. Intersphincteric perianal fistula with concomitant abscess cavity at 11 o’clock in the anal canal (yellow circle). (**b**) Sagittal plane, tridimensional reconstruction. Submucosal perianal fistula on the posterior wall (6 o’clock) of the anal canal (yellow circle).

**Table 1 diagnostics-13-02851-t001:** Demographic characteristics and information about the perianal fistulas for the total sample and the two imaging control groups.

Characteristics	*N* = 72 ^1^	MRI, *N* = 36 ^1^	MRI + 3D-EAUS, *N* = 36 ^1^	*p*-Value ^2^
**Gender**				0.8
**Male**	55 (76%)	28 (78%)	27 (75%)	
**Female**	17 (24%)	8 (22%)	9 (25%)	
**Age**	54 (45, 62)	56 (44, 61)	53 (45, 63)	>0.9
**Height**	1.72 (1.69, 1.78)	1.74 (1.70, 1.79)	1.71 (1.67, 1.76)	0.2
**Weight**	85 (75, 98)	85 (75, 94)	85 (75, 100)	>0.9
**BMI**	28.6 (25.6, 32.0)	26.1 (24.7, 29.9)	29.7 (26.5, 32.9)	0.2
**Primary Fistula**				0.8
**Yes**	17 (24%)	9 (25%)	8 (22%)	
**No**	55 (76%)	27 (75%)	28 (78%)	
**History of Abscess**				0.093
**Yes**	43 (60%)	18 (50%)	25 (69%)	
**No**	29 (40%)	18 (50%)	11 (31%)	
**Fistula Recurrence**				0.4
**Yes**	13 (18%)	5 (14%)	8 (22%)	
**No**	59 (82%)	31 (86%)	28 (78%)	
**Synchronous Treatment (Fistula and abscess)**				0.003
**Yes**	14 (19%)	12 (33%)	2 (5.6%)	
**No**	58 (81%)	24 (67%)	34 (94%)	
**Fistula Type**				0.8
**Simple**	39 (54%)	19 (53%)	20 (56%)	
**Complex**	33 (46%)	17 (47%)	16 (44%)	

^1^ *n* (%); Median (IQR); ^2^ Pearson’s chi-squared test; Wilcoxon rank sum test; Fisher’s exact test.

**Table 2 diagnostics-13-02851-t002:** Frequencies of fistula types that were detected in the two imaging control groups and intraoperatively.

Classification	MRI Group				MRI + 3D-EAUS Group					
	MRI		Intraoperative		MRI		3D-EAUS		Intraoperative	
	N	%	N	%	N	%	N	%	N	%
**Transsphincteric**	9	22	15	36.6	23	50	24	52.2	22	47.8
**Intersphincteric**	25	61	17	41.5	7	15.2	4	8.7	6	13
**Supralevator**	-	-	-	-	2	4.3	2	4.3	3	6.5
**Extrasphincteric**	3	7.3	1	2.4	1	2.2	2	4.3	1	2.2
**Submucosal**	2	4.9	5	12.2	6	13	2	4.3	7	15.2
**Rectovaginal**	1	2.4	1	2.4	ΝA	ΝA	1	2.2	2	4.3
**Perineal**	1	2.4	1	2.4	2	4.3	1	2.2	2	4.3
**No Finding**	ΝA	ΝA	1	2.4	5	10.9	10	21.7	3	6.5

**Table 3 diagnostics-13-02851-t003:** Diagnostic accuracy metrics in the two imaging control groups.

Classification	SN_MRI_	SΝ_adj_	SP_MRI_	SP_adj_	PPV_MRI_	PPV_adj_	NPV_MRI_	NPV_adj_	BA_MRI_	BA_adj_
**Transsphincteric**	0.53	1.00	0.96	0.63	0.89	0.71	0.78	1.00	0.75	0.81
**Intersphincteric**	1.00	1.00	0.66	0.93	0.33	0.67	1.00	1.00	0.83	0.96
**Supralevator**	NA	1.00	NA	1.00	NA	1.00	NA	1.00	NA	1.00
**Extrasphincteric**	1.00	1.00	0.95	0.98	0.33	0.50	0.98	1.00	0.975	0.99
**Submucosal**	0.40	0.71	1.00	0.95	1.00	0.71	1.00	0.95	0.7	0.83
**Rectovaginal**	1.00	0.50	1.00	1.00	0.68	1.00	1.00	0.98	1.00	0.75
**Perineal**	1.00	1.00	1.00	1.00	1.00	1.00	1.00	1.00	1.00	1.00
**No Finding**	0.00	0.67	1.00	0.70	1.00	0.13	0.92	0.97	0.5	0.68

SN_MRI_: sensitivity in the MRI-only group; SN_adj_: sensitivity in the combined MRI-3DEAUS group; SP_MRI_: specificity in the MRI-only group; SP_adj_: specificity in the combined MRI-3DEAUS group; PPV_MRI_: positive predictive value in the MRI-only group; PPV_adj_: positive predictive value in the combined MRI-3DEAUS group; NPV_MRI_: negative predictive value in the MRI-only group; NPV_adj_: negative predictive value in the combined MRI-3DEAUS group; BA_MRI_: balanced accuracy in the MRI-only group; BA_adj_: balanced accuracy in the combined MRI-3DEAUS group.

**Table 4 diagnostics-13-02851-t004:** Number of cases with a perianal fistula and a concomitant abscess cavity in both imaging control groups, preoperatively and intraoperatively.

Concomitant Abscess Cavity	MRI, *N* = 42 ^1^	MRI + 3D EAUS, *N* = 46 ^1^	*p*-Value ^2^
**Preoperative MRI**			0.008
Yes	16 (39%)	6 (14%)	
No	25 (61%)	38 (86%)	
Missing Values	1	2	
**Preoperative 3D-EAUS**			
Yes		5 (12%)	
No		35 (88%)	
Missing Values		6	
**Intraoperative Findings**			0.063
Yes	15 (38%)	8 (19%)	
No	25 (62%)	34 (81%)	
Missing Values	2	4	

^1^ *n* (%); ^2^ Pearson’s chi-squared test; Fisher’s exact test.

**Table 5 diagnostics-13-02851-t005:** Number of cases with a perianal fistula and a secondary fistulous tract in both imaging control groups, preoperatively and intraoperatively.

Secondary Fistulous Tract	MRI, *N* = 42 ^1^	MRI + 3D-EAUS, *N* = 46 ^1^	*p*-Value *^2^*
**Preoperative MRI**			0.044
Yes	5 (12%)	13 (30%)	
No	37 (88%)	31 (70%)	
Missing Values	0	2	
**Preoperative 3D-EAUS**			
Yes		7 (18%)	
No		33 (82%)	
Missing Values		6	
**Intraoperative Findings**			0.043
Yes	3 (7.5%)	10 (24%)	
No	37 (92%)	32 (76%)	
Missing Values	2	4	

^1^ *n* (%); ^2^ Pearson’s chi-squared test; Fisher’s exact test.

**Table 6 diagnostics-13-02851-t006:** Clockwise location of the internal opening in the anal canal in both imaging control groups, preoperatively and intraoperatively.

Inner Orifice Location	MRI, *N* = 42 ^1^	MRI + 3D EAUS, *N* = 46 ^1^	*p*-Value ^2^
**Preoperative MRI**			0.14
Six O’Clock	23 (55%)	15 (34%)	
Other Position	16 (38%)	22 (50%)	
No Inner Orifice	3 (7.1%)	7 (16%)	
Missing Values	0	2	
**Preoperative 3D-EAUS**			
Six O’Clock		19 (49%)	
Other Position		17 (44%)	
No Inner Orifice		3 (7.7%)	
Missing Values		7	
**Intraoperative Location**			0.3
Six O’Clock	25 (62%)	19 (46%)	
Other Position	13 (32%)	20 (49%)	
No Inner Orifice	2 (5.0%)	2 (4.9%)	
Missing Values	2	5	

^1^ *n* (%); ^2^ Fisher’s exact test.

**Table 7 diagnostics-13-02851-t007:** Secondary outcomes at six months in MRI group and MRI + 3D-EAUS group.

Characteristics	*N* = 72 ^1^	MRI, *N* = 36 ^1^	MRI + 3D-EAUS, *N* = 36 ^1^	*p*-Value ^2^
**Closure at Six Months**				
**Yes**	72 (100%)	36 (100%)	36 (100%)	
**Fecal Incontinence (Wexner Scale)**				>0.9
**No Symptoms**	67 (93%)	33 (91.7%)	34 (94.4%)	
**Mild Incontinence, Wexner = 3**	3 (4.2%)	2 (5.5%)	1 (2.8%)	
**Mild Incontinence, Wexner = 4**	2 (2.8%)	1 (2.8%)	1 (2.8%)	
**Repeat Surgery**				0.2
**Yes**	35 (49%)	15 (42%)	20 (56%)	
**No**	37 (51%)	21 (58%)	16 (44%)	

^1^ *n* (%); ^2^ Fisher’s exact test; Pearson’s chi-squared test.

**Table 8 diagnostics-13-02851-t008:** Secondary outcomes at six months in patients with simple and complex perianal fistulas.

Characteristics	*N* = 72 ^1^	Simple, *N* = 39 ^1^	Complex, *N* = 33 ^1^	*p*-Value *^2^*
**Closure at Six Months**				
Yes	72 (100%)	39 (100%)	33 (100%)	
**Fecal Incontinence (Wexner Scale)**				>0.9
No Symptoms	67 (93%)	36 (92%)	31 (94%)	
Mild Incontinence, Wexner = 3	3 (4.2%)	2 (5.3%)	1 (3.0%)	
Mild Incontinence, Wexner = 4	2 (2.8%)	1 (2.6%)	1 (3.0%)	
**Repeat Surgery**				0.019
Yes	35 (49%)	14 (36%)	21 (64%)	
No	37 (51%)	25 (64%)	12 (36%)	

^1^ *n* (%); ^2^ Fisher’s exact test; Pearson’s chi-squared test.

## Data Availability

Available upon request.

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
