# Peer review of "Preoperative Assessment of Perianal Fistulas with Combined Magnetic Resonance and Tridimensional Endoanal Ultrasound: A Prospective Study"

_diagnostics, 2023, doi:10.3390/diagnostics13172851_

Round 1

Reviewer 1 Report (Previous Reviewer 2)

Dear Author;

Thank you for the revisions. It is my opinion that the acceptance of the manuscript is appropriate.

Best regards.

Author Response

Reviewer 2 Report (New Reviewer)

Dear Authors,

The paper is very well written and the topic interesting.

Although the introduction part could be more detailed in describing the pathology and the references could be more up to date.

The results part is very good, although the nr of patients is limited.
I congratulate you for the 5 cases of mild anal incontinence. This is a very important aspect of this pathology.

Kind regards

Few english editing is necessary.

Author Response

This manuscript is a resubmission of an earlier submission. The following is a list of the peer review reports and author responses from that submission.

Round 1

Reviewer 1 Report

I congratulate the authors for a qualitative and beautifully written paper. I enjoyed reading it. The introduction and discussions provide all the necessary context and literature review. The methodology and results are rigorously presented. The conclusions are well sustained. 

Before acceptance, I only have two minor comments:

- Considering that synchronous treatment (fistula and abscess) and the presence of secondary fistulous tract, significantly differed between the groups, these may act as confounding factors. However, the methodology section does not mention about the adjustment for confounding factors or how it was done.

- It would be equally interesting for the reader to also know whether there were significant differences between the type of treatment between the two groups. I was not able to identify this information. Differences among treatments may also act as confounding factors when assessing outcomes.

The paper is scientifically sound and proves qualitative and responsible writing. Congratulations!

Reviewer 2 Report

Dear Author;

1.Both modalities were compared with studies performed by a single evaluator. I think that this situation reduces the reliability of the article. I think that the study should be redesigned by giving at least two evaluator and kappa values for each modality.

2. In a study on Topic Imaging, why surgical methods are discussed in the discussion.

3. Although the MRI was evaluated by the radiologist, there was no author from the radiology department among the authors. For the objectivity of the article, it is obligatory to contribute to the manuscript from the radiology department.

Minor editing of English language required.

Reviewer 3 Report

The manuscript was well written.

Minor comments can be provided

1. IRB protocol number to be provided?

2. It would be interesting to see if there are any race difference between the patients and that could be included in table 1.

3. As indicated in the limitations the small number of patients in the study.
